# Rainfall Projections from Coupled Model Intercomparison Project Phase 6 in the Volta River Basin: Implications on Achieving Sustainable Development

**Sam-Quarcoo Dotse** [1], **Isaac Larbi** [1,*], **Andrew Manoba Limantol** [1], **Peter Asare-Nuamah** [1],
**Louis Kusi Frimpong** [1], **Abdul-Rauf Malimanga Alhassan** [1], **Solomon Sarpong** [2], **Emmanuel Angmor** [1]
**and Angela Kyerewaa Ayisi-Addo** [1]

[1] School of Sustainable Development, University of Environment and Sustainable Development, Somanya 00233, Ghana
[2] School of Natural and Environmental Science, University of Environment and Sustainable Development, Somanya 00233, Ghana
[*] Correspondence: ilarbi@uesd.edu.gh

**Abstract:** Climate change has become a global issue, not only because it affects the intensity and frequency of rainfall but also because it impacts the economic development of regions whose economies heavily rely on rainfall, such as the West African region. Hence, the need for this study, which is aimed at understanding how rainfall may change in the future over the Sahel, Savannah, and coastal zones of the Volta River Basin (VRB). The trends and changes in rainfall between 2021–2050 and 1985–2014 under the Shared Socioeconomic Pathway (SSP2-4.5 and SSP5-8.5) scenarios were analyzed after evaluating the performance of three climate models from the Coupled Model Intercomparison Project Phase 6 (CMIP6) using Climate Hazards Group InfraRed Precipitation with Station data (CHIRPS) as observation. The results show, in general, a relatively high correlation and low spatial biases for rainfall (r > 0.91, −20% < Pbias < 20%) over the entire Volta Basin for the models' ensemble mean. An increasing trend and projected increase in annual rainfall under the SSP2-4.5 scenario is 6.0% (Sahel), 7.3% (Savannah), and 2.6% (VRB), but a decrease of 1.1% in the coastal zone. Similarly, under SSP5-8.5, the annual rainfall is projected to increase by 32.5% (Sahel), +22.8% (Savannah), 23.0% (coastal), and 24.9% (VRB), with the increase being more pronounced under SSP5-8.5 compared to the SSP2-4.5 scenario. The findings of the study would be useful for planning and designing climate change adaptation measures to achieve sustainable development at the VRB.

**Keywords:** climate change; trend analysis; climate models; climate scenarios; Volta Basin

## 1. Introduction

Rainfall is considered one of the most important climate variables and an essential component of the hydrological cycle, as it is the key determinant of all water resources across the world [1]. Since pre-industrialization, rainfall has been characterized by natural variability, both temporally and spatially. However, the onset of industrialization has resulted in the emission of an overwhelming quantum of greenhouse gases (GHGs) into the atmosphere. This has altered the natural dynamics of the global climatic system leading to worsening variability and change in climate variables, as evidenced by frequent unpredictable seasonal and annual rainfall in recent times across the globe [2]. The consequences of worsening rainfall variability and change are being felt in many parts of the world and are expected to worsen in the future [3].

Over the past decades, several studies have examined the climate, its variability, and change at the global, regional, sub-regional, and basin scales [4–6]. Despite the fact that most previous studies relied on different resolutions of climate models, their findings were largely consistent and coherent (i.e., many parts of the world are experiencing rising rainfall

variability and change, as well as the associated socioeconomic risks). For instance, using outputs from an ensemble of 21 global climate models (GCMs), Herrero et al. [7] reported an expected increase in the mean annual rainfall across parts of East Africa. The fifth Assessment Report of the Intergovernmental Panel on Climate Change (IPCC) [8] observed that, even though there was difficulty concluding on the annual rainfall trends in most parts of Africa over the past century due to insufficient observational data, research predicted variability in annual rainfall with a marginal delay in the onset of the rainy season toward the end of the 21st century.

In West Africa, high rainfall variability has affected the hydrological cycle, with a rippling effect on the water resources of most river basins [9,10]. Studies have assessed the impacts of climate change in the sub-region, but insufficient observational data have restricted understanding of the scope, predictions, and projections of such studies as far as the trans-boundary river basins are concerned. For instance, Nouaceur and Murarescu [11] conducted analysis of rainfall variability and trend over the Sahelian part of West Africa and observed a resumption of rainfall in the recent years following a long Sahelian drought, but the scope of the study was limited to three countries (i.e., Senegal, Mauritania, and Burkina Faso). The Volta River Basin (VRB) is an important trans-boundary basin shared by six riparian states (Benin, Burkina Faso, Côte d'Ivoire, Mali, Togo, and Ghana) in West Africa. Due to the high vulnerability of the basin to climate change, any changes in the climate variables, particularly rainfall, could lead to changes in runoff generation and other components of the water balance, which can affect the lives and livelihoods of the agrarian communities in and around the basin.

Rainfall projections are important for developing appropriate adaptation measures to minimize the impacts on society and the environment. Global (GCMs) and regional (RCMs) climate models help in understanding the past climate and projecting future changes. With their continuous development, climate scenarios simulated by GCMs in the various phases of the Coupled Model Intercomparison Project (CMIP) have become a central element of national and international assessments of climate change [10,11]. The recently introduced state-of-the-art CMIP Phase 6 (CMIP6) is made up of models with finer spatial resolutions, extended historical runs, improved parameterizations for cloud microphysical processes, and the inclusion of additional processes and components in the Earth system [11,12]. However, the use of CMIP models over a region requires a systematic and comprehensive assessment of the GCMs' simulation performances at a regional scale [11]. Recently, Agyekum et al. [13] assessed the performance of CMIP5 models in simulating present-day climatology (1950–2004) precipitation over the basin and observed that the models' performances are dependent on the simulation of features that influence the distribution of precipitation. Dembélé et al. [14] also performed a comprehensive evaluation of the impacts of climate change on water resources in the basin using CMIP5 models. They observed contrasting dynamics in the seasonality of rainfall, depending on the selected greenhouse gas emission scenarios and the future projection periods. In Yeboah et al. [15], climate change projections in the Volta Basin using the CORDEX-Africa climate simulations under two emission scenarios of the Representative Concentration Pathways (RCPs), namely RCPs (4.5 and 8.5) of the CMIP5 models, have been assessed. The results showed that the precipitation pattern of the Volta Basin is decreasing with time under RCP 4.5 and 8.5, along with a more frequent and intense dry period.

To the best of the authors' knowledge, there is limited research, if any, employing the recently introduced CMIP6 models for assessing rainfall patterns in the Volta Basin. Even though the study by Ajibola et al. [16] covered the entire West Africa and included the basin, they only evaluated the historical performance of the High-Resolution Model Intercomparison Project (HighResMIP) simulations within the framework of the CMIP6 models. Information on future rainfall projections, which is relevant in designing robust policies for the management of the natural resources of the basin, is still lacking. Moreover, studies on observed and anticipated climate change and variability using models have also

become important in making policy decisions regarding adaptation and mitigation [17]. However, studies on climate change modeling are not often discussed within the context of sustainable development, especially how such studies can help to achieve the sustainable development goals. This study therefore seeks to move the discussion of the findings within the sustainable development discourse. Given that the effects of climate change are felt at many spatial scales and in various societal sectors, it raises important questions about how countries, including those in West Africa, can advance toward meeting sustainable development goals (SDGs). In this study, we discuss the implications of the findings with respect to the attainment of the SGDs in the six riparian states.

The aim of the present study is to fill the identified gap by assessing the changes in rainfall under the Shared Socioeconomic Pathways (SSP2-4.5 and SSP5-8.5) emission scenarios over the Sahel, Savannah, and Coastal Zones of the VRB using climate outputs from CMIP6 models. Analysis of the past, present, and future rainfall variability and change using CMIP6 models will improve our understanding of the future climate of the basin and allow for more reliable prediction of climate risks that are associated with local weather phenomena, such as droughts and floods [18].

## 2. Materials and Methods

### 2.1. Study Area Description

The VRB (Figure 1), located between longitudes $5°30'30''$ W and $2°0'30''$ E, and latitudes $6°0'30''$ N and $15°0'30''$ N, with an estimated surface area of about 414,000 km$^2$, is shared by six riparian states (Benin, Burkina Faso, Côte d'Ivoire, Mali, Togo, and Ghana) in West Africa. The climate of the basin is controlled by the movement and interactions of the Inter-Tropical Discontinuity (ITD) and the associated West African Monsoon [19]. Located within three climatic zones, the tropical climate, the humid south, and the tropical transition zone, the basin is characterized by a bimodal rainfall regime in the south and a unimodal one in the north. The basin has high annual rainfall variation, which ranges from 300 mm (North) to 1700 mm (South), and temperatures ranging from 25.0 °C (South) to 30.0 °C (North) (Figure 1C). The Guinea Coast is characterized by a bimodal pattern of precipitation (a maximum peak in June and a second one in September), while the Savannah and Sahel regions are characterized by a unimodal pattern (the maximum peak in August in both cases). The topography at the basin is uniformly flat, with about 80% of the area between $-1$ m and 400 m and steeper in some portions in the eastern and northwestern parts (Figure 1B). In the VRB, an estimated population of about 19 million depends directly or indirectly on the basin for their water supply and agricultural activities.

### 2.2. Observation and CMIP6 Models Data

Daily rainfall data for the period 1985–2014 obtained from satellite-climate products were used in this study. Studies such as Paeth et al. [20] and Larbi et al. [21] have indicated that the availability of high-quality datasets, especially in the case of precipitation over West Africa, is problematic. As a result, gridded observational open-source datasets have become a practicable option for model validation in the region. The rainfall data were from Climate Hazards Group Infrared Precipitation with Stations (CHIRPS) at 5 km grid resolution [22]. CHIRPS data have been frequently used as observation data over West Africa by several studies [23,24]. Other studies, such as Larbi et al. [21], have demonstrated the robustness of the CHIRPS data in reproducing observed rainfall within the West African region. The preceding studies have thus given confidence to the use of this particular dataset, though potential uncertainties remain. Notwithstanding, validation of the CHIRPS data over the three subdomains (Guinea Coast, Savanna, and Sahel) within the Volta Basin at the monthly scale (Figure A1) was further performed using station data before the datasets were used as observations.

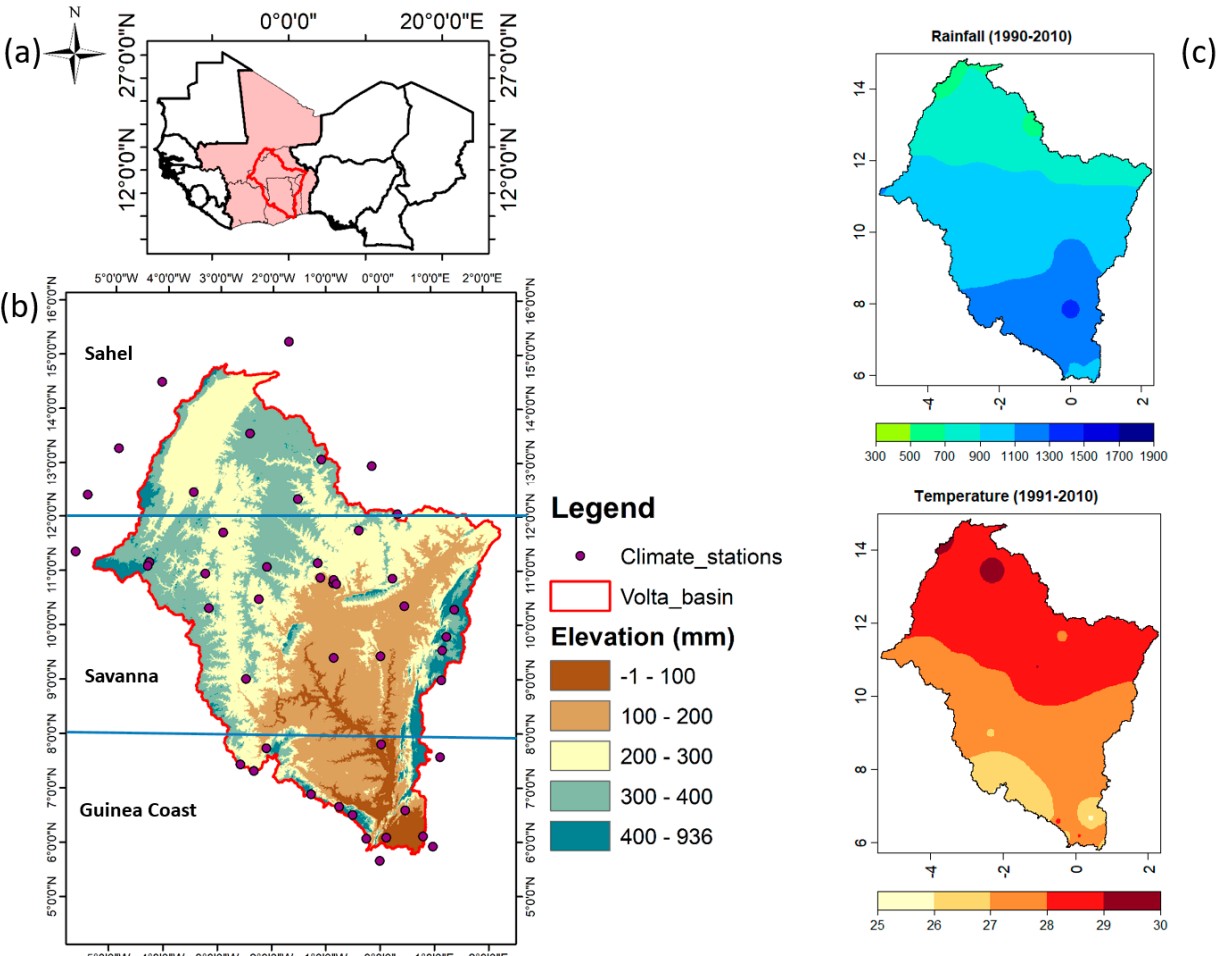

**Figure 1.** The Volta Basin showing: (**a**) the basin within West Africa; (**b**) elevation, climate stations distribution, and domain designated as Guinea Coast (6° N–8 °N), Savanna (8° N–12° N) and Sahel (12° N–15° N); and (**c**) spatial distribution of rainfall and temperature for the period 1991–2010.

Three (3) GCMs' (Table 1) rainfall outputs at 100 km spatial resolution from the Coupled Model Intercomparison Project (CMIP6) simulations for the historical (1985–2014) and future (2021–2050) were used. The GCMs are from the Oceanic Global Climate Model (OGCM) and are the latest phase of a coordinated effort by modeling groups across the globe. The models were selected by considering their equilibrium climate sensitivity (ECS) values, which are the expected long-term warming after a doubling of atmospheric carbon dioxide ($CO_2$) concentrations. The selected GCMs have relatively low equilibrium climate sensitivity in the range from 1.9 to 3.0, which is consistent with the fifth Assessment Report (AR5) range. The new CMIP6 climate change scenarios are based on five Shared Socioeconomic Pathways (SSPs) combined with different forcing levels (i.e., Representative Concentration Pathways, RCPs) to form eight main scenarios.

**Table 1.** Details of CMIP6 climate models used in this study.

| Model | Institute | Reference |
|---|---|---|
| BCC-CSM2-MR | Beijing Climate Centre (BCC) and China Meteorological Administration (CMA), China | Wu et al. [25] |
| NorESM2-MM | Norwegian Climate Centre, Norway | Swart et al. [26] |
| MPI-ESM1-2-HR | Max Planck Institute, Germany | Gutjahr et al. [27] |

In this study, an emission scenario that combines SSP2 (i.e., a central pathway in which trends continue their historical patterns without substantial deviations) and RCP4.5, hereafter called the SSP 2-4.5 scenario, and the one that combines SSP5 (i.e., energy intensive, fossil-based economy) with RCP8.5, hereafter called the SSP 5-8.5 scenario, were used. The SSP 2-4.5 scenario combines intermediate societal vulnerability with an intermediate forcing level and a warming in the range from 2.1 to 4.3 °C, while SSP5-8.5 is at the high end of future pathways with a warming in the range from 3.8 to 7.4 °C. These scenarios were selected in order to understand the future rainfall and temperature patterns under very high $CO_2$ emissions and mitigation challenges (i.e., SSP5-8.5) and medium $CO_2$ emissions and adaptation challenges (i.e., SSP 2-4.5). The data are archived at the Earth System Grid Federation (ESGF) website under CMIP6 [28].

### 2.3. Performance Evaluation of CMIP6 Models

The performances of the CMIP6 models in simulating the observed rainfall over the entire Volta Basin and for the three sub-regions (Guinea Coast, Savannah, and Sahel) were evaluated at monthly and annual scales for the period 1985–2014 using the satellite observational data. The annual cycle of the monthly mean analysis was used to assess how well the CMIP6 models reproduce the bimodal rainfall (Guinea Coast) and unimodal rainfall (Sahel and Savanah) patterns over the Volta Basin. The Taylor diagram was used to further evaluate the performance of the CMIP 6 models at the monthly scale using indicators such as root mean square error (RMSE), normalized standard deviation ($\sigma$) and Pearson correlation coefficient (r) [29]. The RMSE, $\sigma$, and r represent the temporal errors in the models, the temporal pattern, and the temporal variability, respectively. At the spatial scale, the biases between the models and the observations were also estimated. The biases describe the relative systematic error associated with the CMIP6 models' data, where a positive or negative value indicates overestimation and underestimation of the observed rainfall data, respectively.

### 2.4. Rainfall Changes, Trends and Uncertainty Analysis

The flowchart for the study is presented in Figure 2. Rainfall analysis for the past (1985–2014) and future (2021–2050) periods was performed at both monthly and annual scales using the multi-model mean (MME) of the three CMIP6 models. The selection of MME was to minimize the uncertainties of future climate projections [30]. The projected absolute changes in the mean annual rainfall at the temporal and spatial scales were estimated by determining the difference between the mean historical (1985–2014) climate and the future (2021–2050) climate under the different climate scenarios. The *t*-test was used to determine whether the obtained changes on the temporal scale were significant or not. Furthermore, the mean annual cycle of the monthly rainfall from 1985–2014 was compared with the selected scenarios. The spatial distributions of the changes in the past and future rainfall were also analyzed with the R-software version 3.3.0.1959 (available at cran.r-project.org/mirrors.html (accessed on 5 October 2022)) using the Inverse Distance Weighting (IDW) technique. The IDW interpolation technique is based on the concept of distance weighting, which is used to estimate the unknown spatial rainfall data from the known data of sites that are adjacent to the unknown site [31]. The choice of the IDW compared to other interpolation techniques is its usefulness when the distribution of the estimated parameters is not a normal distribution [32], as in this study. The deterministic IDW interpolation technique has also been demonstrated to perform well in spatial rainfall distribution [21,33]. Detailed information on IDW is found in the study by Feng-Wen and Chen-Wuing [32].

Uncertainty levels associated with GCM outputs are crucial in climate change assessment. The violin plot was used to quantify the inter-modal spread of the projected changes in rainfall in each zone and the entire basin. It was also used to gain some insights about the associated uncertainties. The violin plot is basically a boxplot with the added benefit of an overlaid distribution plot of the data (i.e., it shows how a dataset varies along one

variable by combining a boxplot with a probability density function). The probability density function (PDF) is a smoothed histogram used to show the shape of the dataset, with a wider PDF indicating that the value occurs more frequently. A narrower PDF indicates that the value occurs less frequently. The interquartile range of a boxplot expresses how scattered the data are, with a higher value indicating a higher level of variability and, thus, a higher level of uncertainty, and vice versa.

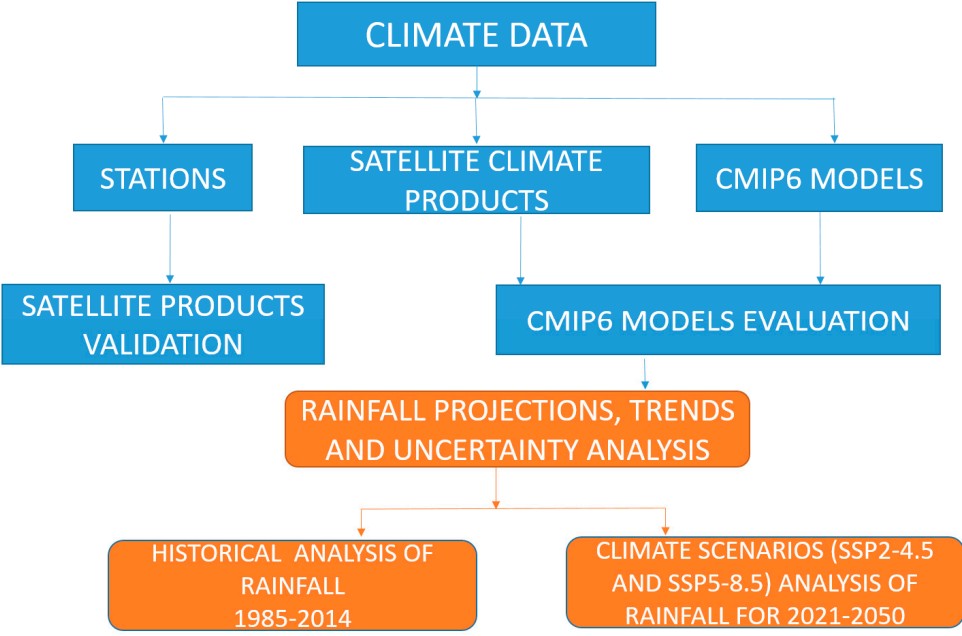

**Figure 2.** Flowchart for the study.

To obtain an idea about the trends in the rainfall in the entire Volta Basin and the three zones for the period 1985–2050, the non-parametric Mann–Kendall (MK) test was computed at a 5% significant level. The MK test has been utilized in several studies (e.g., Okafor et al. [34]; Larbi et al. [21]; Nyembo et al. [35]) and proven to be suitable for non-normally distributed hydro-meteorological data. The MK test assumes a null hypothesis (Ho) that there is no trend, which is tested against the alternative hypothesis (H1) of the presence of a trend [36]. Positive and negative values of MK test statistics (Z) indicate upward and downward trends, respectively. Absolute Z values greater than or equal to 1.28, 1.68, and 2.32 indicate significance levels of 90%, 95%, and 99%, respectively. The magnitude of the trend was estimated using Theil-Sen's estimator.

## 3. Results

### 3.1. CMIP6 Models Perfomance Evaluation for Rainfall

#### 3.1.1. Temporal Distribution

The temporal distribution of the annual cycle of the mean monthly rainfall show that all the models were able to capture the general rainfall pattern in each zone (i.e., Sahel, Savannah, Guinea Coast) and over the Volta Basin with some discrepancies (Figure 3). For example, in the Sahel region, NorESM2-MM overestimates the amount of rainfall during the peak season, while MPI-ESM1-2-HR, BCC-CSM2-MR, and the ensemble underestimate the rainfall values during the peak season. For the Guinea Coast, the bimodal rainfall patterns were captured by BCC-CSM2-MR and NorESM2-MM with some overestimation and underestimation, respectively.

Presented in Figure 4, the models' performances are evaluated at the monthly scale using Taylor's diagram based on the Pearson correlation coefficient (r), root mean square error (RMSE), and normalized standard deviation. A higher correlation (r > 0.9) was found

for all the models in the Sahel, Savannah, and entire Volta Basin, while in the Guinea Coast zone, the correlation ranged between 0.7 and 0.9 (Figure 4).

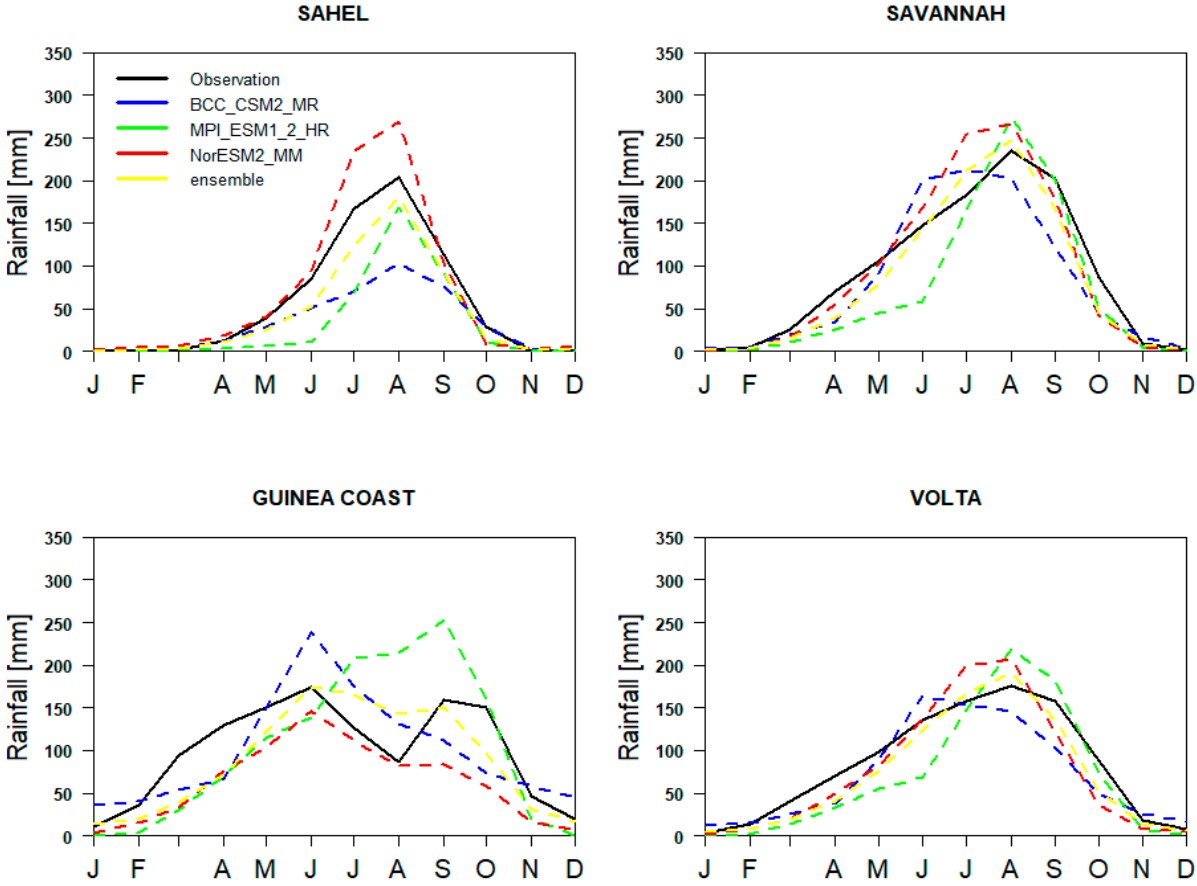

**Figure 3.** Annual cycle of mean monthly rainfall: comparison between observation and CMIP6 models for the period 1985–2014 showing unimodal (Sahel, Savannah and Volta) and bimodal (Guinea Coast) patterns.

With the exception of the Sahel zone, the standard deviation values are below 1.0. In the case of the Sahel zone, the MPI-ESM1-2-HR and BCC-CSM2-MR models have standard deviation values of 1.8 and 3.0, RMSE values of 1.75 and 2.0, respectively.

3.1.2. Spatial Biases in Rainfall Distribution

The spatial pattern of rainfall (Figure 5) shows some biases ranging between −20% and 20%, which cover about 90% of the basin for all the models except BCC-CSM2-MR, which presents biases greater than −20% in the Sahel region. The BCC-CSM2-MR shows underestimation of rainfall, the MPI-ESM1-2-HR shows overestimation, and the NorESM2-MM shows both overestimation (in the north of the basin) and underestimation (in the south of the basin) of rainfall.

*3.2. Rainfall Projections and Trends under SSP2-4.5 and SSP5-8.5 Scenarios*

Under the SSP5-8.5 scenario, the Sahel, Savannah, coastal, and entire Volta Basin would experience an increase in the mean annual and monthly rainfall (Table 2 and Figure 6). Similar projections are expected under SSP2-4.5, with the exception of the coastal zone, which is expected to experience a reduction in both mean annual and monthly rainfall (Figure 6) in the future. A higher increase in annual rainfall under the SSP5-8.5 scenario is projected for Sahel (165.5 mm) and Savannah (219.5 mm) zones compared to SSP2-4.5 for Sahel (30.3 mm) and Savannah (70.5 mm). In the coastal zone, annual rainfall is projected to decrease by 12 mm under the SSP2-4.5 scenario but to increase by 240.2 mm under

SSP5-8.5. In general, the projected changes in mean annual rainfall under SSP2-4.5 were not statistically significant, while the projected changes under SSP5-8.5 were significant at a 95% confidence level.

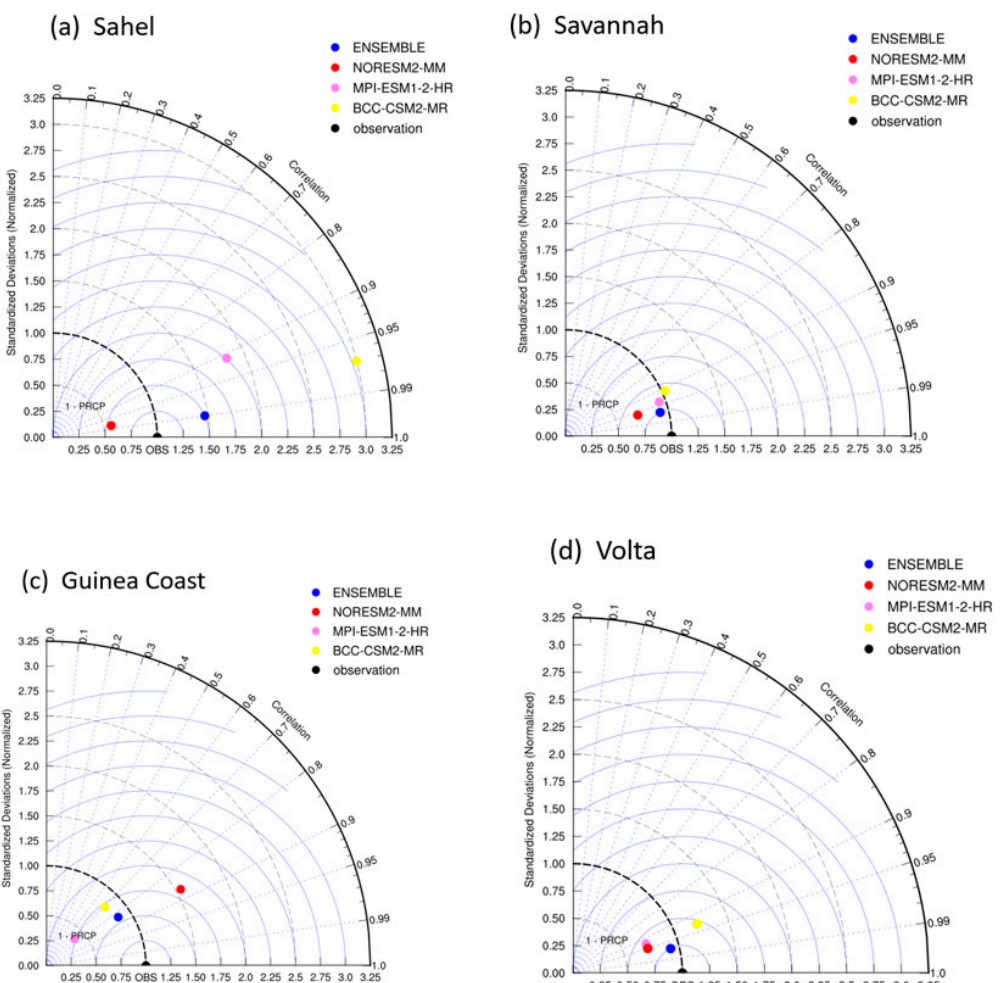

**Figure 4.** Taylors diagram showing statistical comparison (i.e., normalized standard deviation (*x*-axis), correlation and RMSE) at the monthly scale between observation and GCM rainfall over the period of 1985–2014.

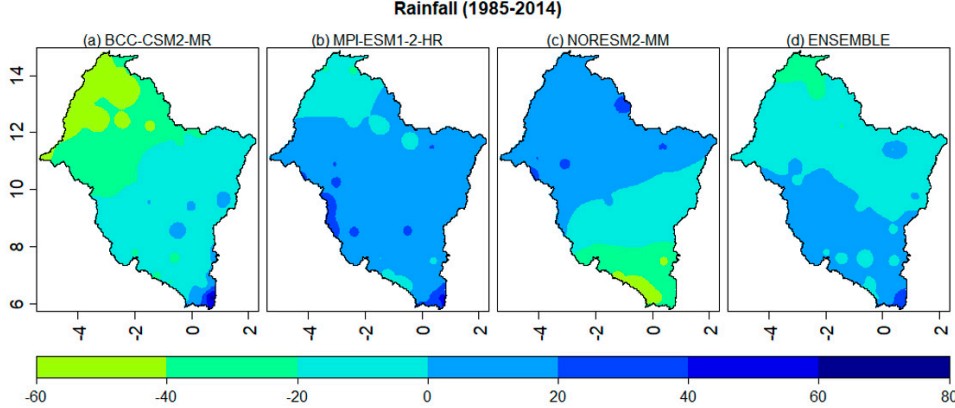

**Figure 5.** Spatial distribution of the biases (PBIAS) in the annual rainfall by the CMIP6 models relative to the observation over the Volta Basin.

**Table 2.** Mean annual rainfall (mm) values from the CMIP 6 model ensemble for the various zones and the Volta Basin for the simulated historical (1985–2014) and the two climate scenarios in the future (2021–2050) period.

| Zones | Historical | SSP2-4.5 | SSP5-8.5 |
|---|---|---|---|
| Sahel | 508.9 | 539.2 (+6.0) | 674.4 (+32.5) * |
| Savannah | 962.7 | 1033.2 (+7.3) | 1182.2 (+22.8) * |
| Coastal | 1043.5 | 1031.5 (−1.1) | 1283.7 (+23.0) * |
| Volta | 838.4 | 860.2 (+2.6) | 1046.9 (+24.9) * |

Note: projected changes (%) in annual rainfall values are in the bracket. * means significant at 95% confidence level.

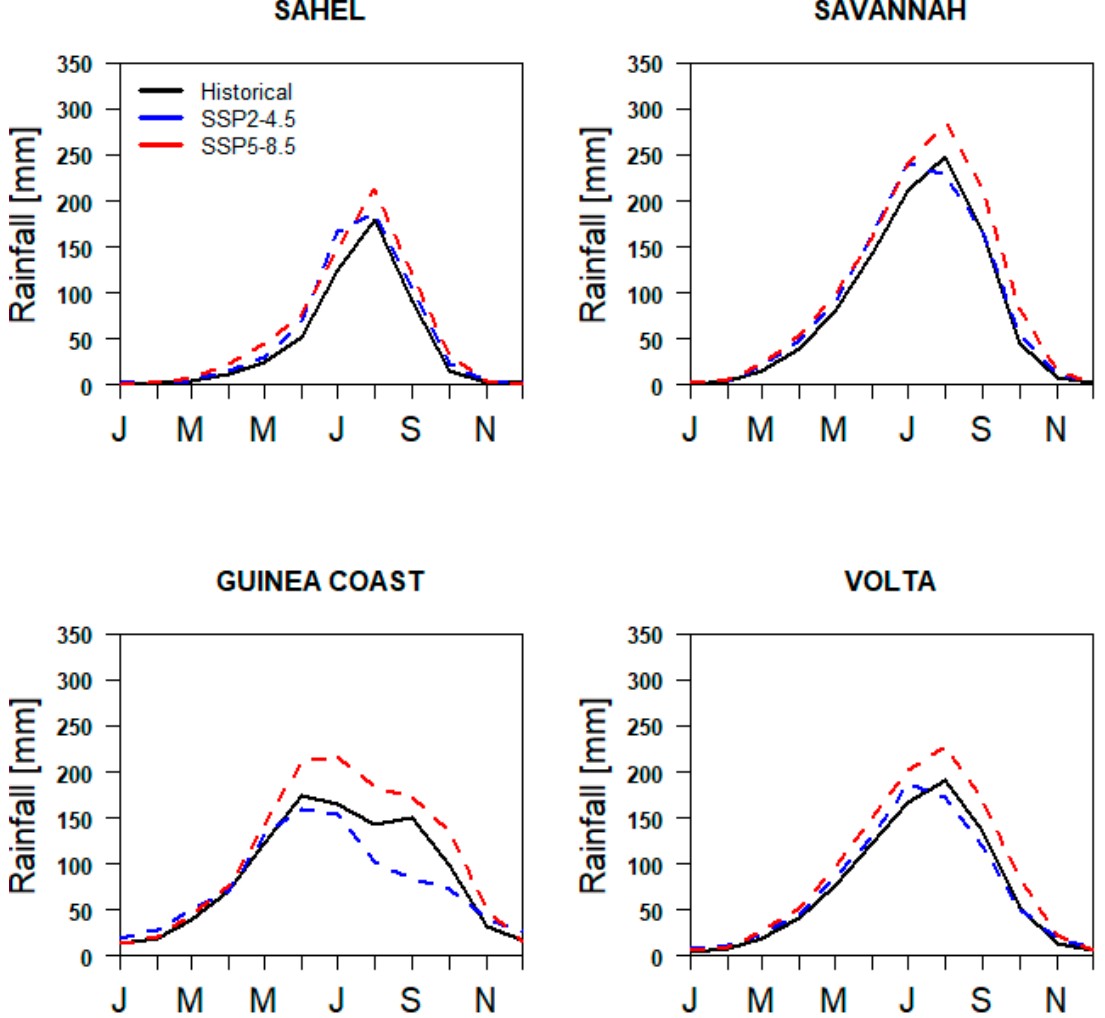

**Figure 6.** Monthly rainfall projections over the Volta Basin for the historical and future under the SSP2.45 and SSP845 climate scenarios.

The spatial distribution of CMIP6 historical, future, and projected changes in rainfall under the SSP2-4.5 and SSP5-8.5 climate scenarios over the Volta Basin is shown in Figure 7. The rainfall pattern in the basin follows a north–south gradient, with an increase in rainfall from the north to the south (Figure 7a). Under the SSP2-4.5 scenario, the rainfall is projected to increase in the Sahel and Savanah zones but decrease in the coastal zone, while under the SSP5-8.5 climate scenario, the result shows a projected increase in rainfall across the Volta Basin (Figure 7b). The spatial increase in rainfall at the basin is found in the Savannah and Sahel zones and is higher under SSP5-8.5 compared to SSP2-4.5 (Figure 7c).

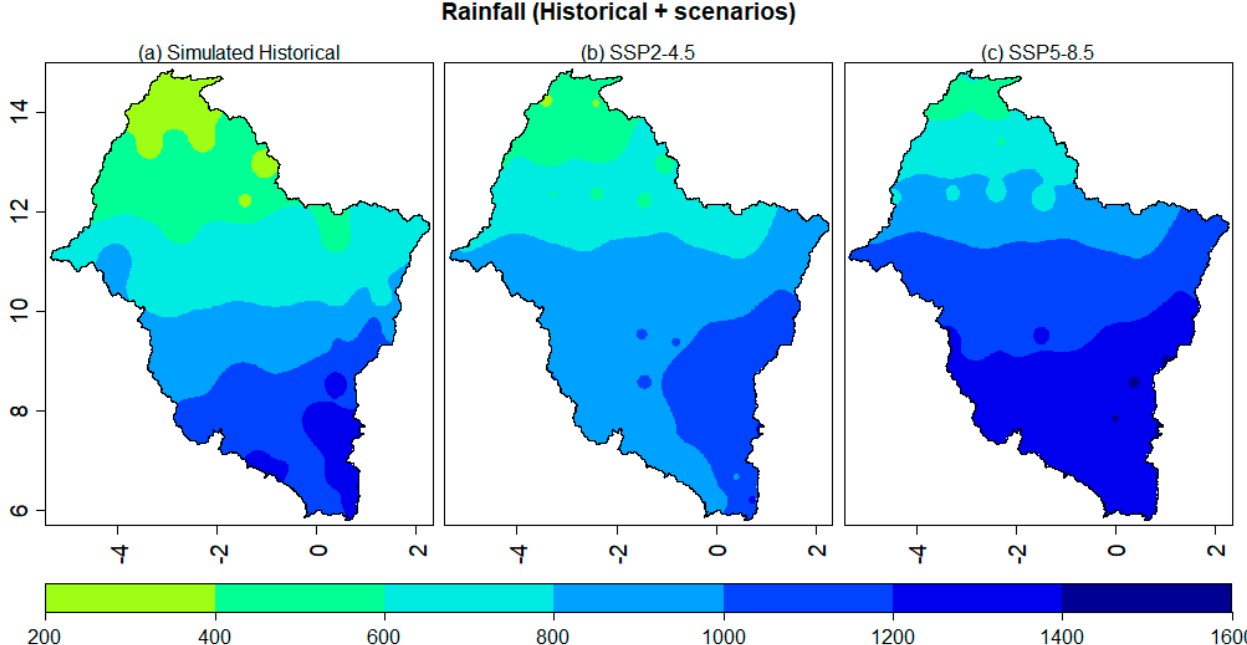

**Figure 7.** Spatial distribution of CMIP 6 historical (**a**), projected (**b**,**c**) and changes (**d**,**e**) in annual rainfall (mm) under the SSP2.45 and SSP5-8.5 climate scenarios over the Volta Basin.

The trend analysis results for the annual rainfall over the Volta Basin from the years 1985 to 2050 under the two scenarios of climate change are presented in Table 3. The results show a statistically increasing trend in rainfall in the Sahel and Savannah zones, while the coastal zone shows a statistically decreasing trend under the SSP2-4.5 scenario. The rainfall trend under SSP5-8.5, however, shows a statistical increase in all the three zones.

### 3.3. Projected Changes in Rainfall and Its Uncertainties Associated with the GCMs

It is evident from Figures 8 and 9 that there are differences in the range of rainfall projection among the three models in the different zones. For example, in the Sahel zone under the SSP2-4.5 scenario, the percentage changes in rainfall are within the range from −38% to 16.6% for NorESM, −35.1% to 31.2% for BCC_CSM2, −34.5% to 76.5% for

MPI_ESM, and −13.6% to 26.4% for the ensemble mean. Similarly, in the Volta Basin, the percentage changes in rainfall are within the range from −30% to 48% for NorESM, −28% to 41% for BCC_CSM2, −16% to 20% for MPI_ESM, and −9.9% to 26% for the ensemble mean. Under the SSP5−8.5 scenario (Figure 8), the changes in rainfall in the Volta Basin are within the range from −13% to 55% for NorESM, −42% to 29% for BCC_CSM2, −17% to 30% for MPI_ESM, and −9% to 30% for the ensemble mean. It can therefore be seen under both scenarios in the Volta Basin that a higher range of the projected changes in rainfall is found in NorESM, followed by BCC_CSM2, with MPI_ESM and the ensemble mean indicating the lowest range of rainfall projections.

**Table 3.** Mann–Kendall trend test (Z) and Sen's slope (Q) estimator results for annual rainfall from 1985–2050 under the SSP2-4.5 and SSP5-8.5 scenarios of climate change.

| Zones | CMIP 6 Historical+ SSP2-4.5 (1985–2050) | | CMIP 6 Historical+ SSP4-8.5 (1985–2050) | |
|---|---|---|---|---|
| | Z | Q | Z | Q |
| Sahel | 3.88 ** | 2.1 | 5.93 ** | 3.8 |
| Savannah | 2.56 ** | 1.5 | 5.91 ** | 4.9 |
| Coastal | −3.08 ** | −3.0 | 5.38 ** | 4.9 |
| Volta Basin | 0.38 | 0.20 | 5.93 ** | 4.4 |

** indicate significance level of the trend at 95%.

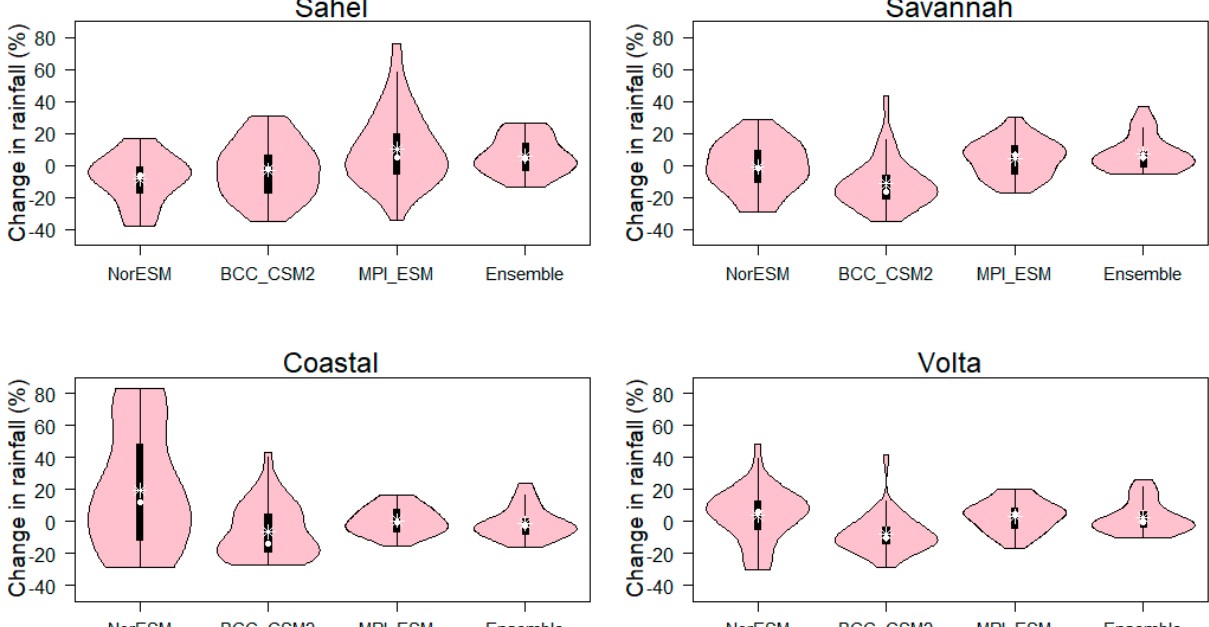

**Figure 8.** Violin plots of the projected changes in annual mean rainfall between the future (2021–2050) and the historical (1985–2014) period by the different GCMs and the ensemble under the SSP2-4.5 scenario for Sahel, Savannah, Guinea Coast, and the Volta Basin.

The uncertainty levels associated with the GCMs' projections, as shown by the interquartile range (i.e., black bar in the center of the violin), is different for the individual models depending on the location (Figure 8). For example, in the coastal zone under the SSP2-4.5 scenario, a very high level of uncertainty (i.e., higher value of IQR) is found in the NorESM (IQR = 18.7%), followed byBCC_CSM2 (IQR = 13.4%), with MPI_ESM indicating the lowest level of uncertainty (IQR = 12.8%) in the rainfall projections. Similarly, in the coastal zone, a very high level of uncertainty (i.e., higher value of IQR) is found in the NorESM (IQR = 81.7%), followed by BCC_CSM2 (IQR = 23.9%) with MPI_ESM indicating the lowest uncertainty (IQR = 14.4%). In general, the ensemble mean has the lowest IQR, with the exception of the Sahel zone, where it appears to be similar to the NorESM, as

shown in Figure 8. Similar to the uncertainty in the models under SSP2-4.5, Figure 9 depicts that in most cases the ensemble mean shows the lowest degree of uncertainty (i.e., shorter IQR) in projecting the changes in rainfall when compared to the individual GCMs under SSP5-8.5 in the Volta Basin.

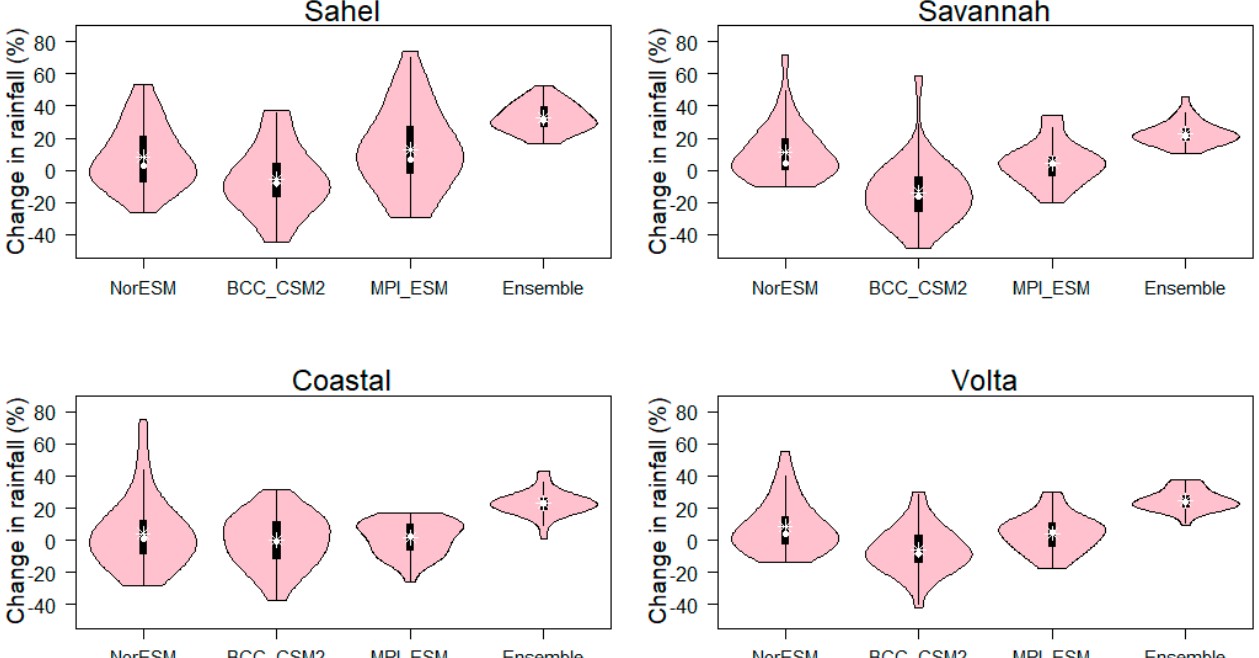

**Figure 9.** Violin plots of the projected changes in annual mean rainfall between the future (2021–2050) and the historical (1985–2014) period by the different GCMs and its ensemble under the SSP5-8.5 scenario for Sahel, Savannah, Guinea Coast, and the Volta Basin.

## 4. Discussion

The present study assessed the future changes in rainfall by first and foremost evaluating the GCMs' ability to mimic the patterns of observed rainfall. As shown in Figure 2, the observation data exhibit a single peak of rainfall over the Sahel and Savannah zones and two peaks of rainfall over the Guinea Coast (i.e., a primary maximum in June and a secondary one in September). These two different rainfall patterns (i.e., unimodal and bimodal) were all captured relatively well by the three individual CMIP6 GCMs and their ensemble. These, together with the overall good values of the statistical indicators used, lend credence to the robustness of the ensemble mean of the three CMIP6 GCMs that were used. This is consistent with the observation by Kim et al. [37] that there has been an improvement in the performance of the CMIP6 GCMs for modeling rainfall in comparison with the CMIP5.

The determination of the biases between the models and the observations is also a key measure of the robustness of GCM outputs. With the exception of the BCC-CSM2-MR model that presents dry biases in rainfall greater than −20% over the Sahel region, the spatial pattern of rainfall shows some biases ranging between −20% and 20% over most parts of the Volta Basin (about 90% coverage) for all the models. What this means is that the models' outputs show a mix of some underestimation, some overestimation, and accurate estimation of the spatial pattern of rainfall over the basin. According to Sood and Smakhtin [29] and Jain et al. [30], the causes of uncertainties that lead to either overestimation or underestimation of values of GCMs can be mainly attributed to: (i) approximations during numerical modeling; (ii) different feedback mechanisms (cloud and solar radiation, greenhouse gases, aerosols, natural and anthropogenic sources, ocean circulation, water vapor and warming, ice and snow albedo); and (iii) physical parameterizations, initializations, and model structures. In the Volta Basin, as further posited by Agyekum et al. [13], the rain-

fall, which is determined mainly by two air masses, namely, the dry, cold "northeasterlies" winds from the Sahara, and the moist, warm "southwesterlies" winds from the Atlantic Ocean, is likely to be overestimated or underestimated by GCMs that do not adequately represent these two air masses but exaggerate their effects. However, in comparison with previous studies (e.g., [13]), the estimated bias range between −20% and 20% obtained by the three CMIP6 GCMs is relatively acceptable. For instance, in the evaluation of CMIP5 GCMs over the Volta Basin, Agyekum et al. [13] showed that the CMIP5 GCMs estimated the observed rainfall over most parts of the basin with a relative bias between −114% and 196%. This shows the relative improvement in CMIP6 GCMs' performance over the Volta Basin. According to Agyekum et al. [13], the multi-model ensemble mean is recommended for use over the Volta Basin because of its ability to estimate comparatively accurate rainfall over the area.

Under the SSP2-4.5 scenario, spatial-temporal variability was found in the rainfall projections. Eyring et al. [10] clarified that, while GCMs generally agree on expected temperature increases across the globe, the issue of where and how rainfall will change, remains diverse among the models. This, according to Jain et al. [30], can be attributed to the high variability in rainfall, physical parameterizations, initializations, and climate model structures. This supports the observed spatial variability of the changes in rainfall at the Volta Basin under the SSP2-4.5 scenario, where rainfall is projected to decrease at the coastal zone but increase at the Sahel and Savannah zones. Our study also predicts a significant increase in rainfall on a spatiotemporal scale under the SSP5-8.5 scenario, which is consistent with the findings of Eyring et al. [10], which indicate that GCMs generally predict a 16–24% increase in heavy rainfall intensity in most parts of the world in the future. The link between changes in rainfall due to climate change and the attainment of the Sustainable Development Goals (SDGs) is a strong one. The rainfall projections in the Volta Basin are expected to have a significant impact on the availability and accessibility of basic human needs such as food and water. In the Volta Basin, agriculture is the dominant economic activity and accounts for 40% of the basin's economic activities. Depending on the scenario considered, our study findings have both positive and negative implications for agriculture. For example, under the SSP2-4.5 scenario, the projected increase in the mean annual rainfall (which is less than 10%) in the Sahel and Savannah zones can significantly improve agricultural productivity (i.e., increase in crop, livestock and fish yields) if well distributed in the growing season there by ensuring food security (i.e., SDG 2), and reduce poverty (i.e., SDG 1). These improvements lead to positive change in the living conditions of farming households. This is consistent with studies that report a positive relationship between rainfall and agriculture [38]. Intuitively, improvement in agricultural production through reliable rainfall will lead to a positive effect on the incomes and livelihoods of farming households where poverty rates are high [39]. However, the very high increase in rainfall under the SSP5-8.5 scenario, if not well distributed in the growing season, may lead to flooding that can cause devastation effects on agriculture and other livelihoods.

Without doubt, poor smallholder farmers in Africa form the majority of the agricultural labor force and contribute to about 80% of food production on the continent [40]. As such, climate-induced poverty, poor nutrition, and food insecurity in vulnerable smallholder farming communities will be disastrous to the economic development of African economies, where agriculture is the major contributor to gross domestic product and foreign income as well as the major employer [41,42]. The tendency of climate change to reduce the contribution of agriculture to the gross domestic product has been highlighted [43]. According to the World Bank [43,44], addressing poverty and food insecurity is instrumental in enhancing sustainable economic growth and development, especially in vulnerable and at-risk regions such as the Volta river basin. Consequently, improvements in agriculture, which is the major economic activity in the basin, will play a key role in boosting the local economies of the riparian states.

On the contrary, the projected decrease in rainfall in the coastal zone under the SSP2-4.5 scenario poses a negative impact on the water resources, and therefore necessitates

innovative methods of water storage for households during the dry seasons [45]. Under the SSP5-8.5 scenario, the very high increase in rainfall at the basin can have a negative impact, such as flooding, thereby affecting agricultural productivity and economic growth in the catchment area in the future. In the basin, experiences in the past have shown that there are occasional erratic rainfall periods that characterize the three zones [46]. Previous studies in Ghana indicate that climate extremes such as floods have resulted in a drastic reduction in the national output of maize (6.3%) and rice (9.3%) [47]. This is problematic as it has serious implications on household food security, as a result of the rising prices of food commodities [48], thereby affecting the purchasing power of poor households who form the majority in developing economies of the Volta Basin. Arndt et al. [42] also contend that climate change is a big blow to developing economies, such as Ghana, as it significantly reduces national welfare, with severe impact among poor households.

The SDG 7 emphasizes the need for states to promote the adoption and use of clean and affordable energy, which is required to drive sustainable development. Fortunately, the Volta Basin serves as an important source of energy, particularly hydropower, which is required for industrialization, economic growth, and development [34]. However, riparian states, as well as Africa as a whole, face significant challenges in ensuring consistent energy supply for economic growth and development. The results of the study show that a future climate scenario of increased rainfall offers a great opportunity for energy supply in the Basin. For instance, in the case of the Akosombo dam in Ghana, this means more water will be received from its headwaters, which will also lead to a maximum functioning of the dam. However, the likelihood of high variability in rainfall patterns (e.g., under SSP5-8.5) can derail energy generation and supply, which is consistent with existing studies [49]. Implicitly, a rise or a decline in rainfall over the Volta Basin has implications for the economic development of riparian states where the basin serves as the major, if not the main, source of hydropower. For instance, regular generation and supply of hydropower, facilitated by an increase in rainfall, will boost the economic activities of households and industries, thereby positively affecting national economic development. On the contrary, a decline in rainfall will affect access to energy, with a significant negative effect on the productivity of households and industries. Thus, there is a need to increase the energy supply mix for countries in the basin to offset any uncertainty.

## 5. Conclusions

Climate change is a major challenge in achieving sustainable development globally, particularly in West Africa. To contribute to our understanding of climate change and how it affects the SDGs, this study assessed the changes in rainfall under different climate change scenarios over the Volta River Basin in West Africa using an ensemble mean of three CMIP-6 models. Annual rainfall is expected to increase significantly over the Sahel zone, Savannah zone, and generally over the entire VRB under the SSP2-4.5 scenario, although it is expected to drop in the coastal zone. Similar to SSP2-4.5, the SSP5-8.5 scenario predicts a significant increase in annual rainfall in the Sahel, Savannah, and coastal zones and generally over the entire VRB, with the increase under SSP5-8.5 being more pronounced than the increase under SSP2-4.5. The findings of the present study using CMIP6 models give an improved understanding of the future rainfall pattern of the basin relative to CMIP5 models, allowing for more reliable prediction of climate risks that are associated with local weather phenomena, such as droughts and floods. The findings, for example, show that the anticipated changes in rainfall patterns in the three zones (Sahel, Savannah, and coastal) necessitate different adaptive capacities to climate change. As an example, in these two zones (the Sahel and the Savannah), coping capacities need to be increased in dealing with flood events due to the higher rainfall projections under the SSP5-8.5 scenario. This may include intensive sensitization and resilient infrastructure by governments and the Volta Basin Authority to minimize the effects of flood events that may occur. Higher-resolution regional climate models (RCMs) would be preferable for rainfall projections, but GCMs were used due to the unavailability of RCMs in the CMIP-6 project. Future

studies using RCMs or more than three GCMs for rainfall projections over the area are recommended. Employing more than three GCMs could help reduce model uncertainty and improve projections.

**Author Contributions:** S.-Q.D., I.L. and A.M.L.; funding acquisition, project administration, conceptualization, methodology, writing—original draft preparation. P.A.-N., L.K.F., A.-R.M.A., E.A., S.S. and A.K.A.-A.; writing—review and editing. All authors have read and agreed to the published version of the manuscript.

**Funding:** This research was funded by African Economic Research Consortium (AERC) under the AERC collaborative research project on Climate Change and Economic Development (CCED), grant number RC21585 and "The APC was funded by AERC".

**Institutional Review Board Statement:** Not applicable.

**Informed Consent Statement:** Not applicable.

**Data Availability Statement:** The data presented in this study are available in this manuscript.

**Acknowledgments:** The authors are grateful to AERC and to also the Climate Computing Community for making freely available the CMIP 6 models.

**Conflicts of Interest:** The authors declare no conflict of interest.

## Appendix A

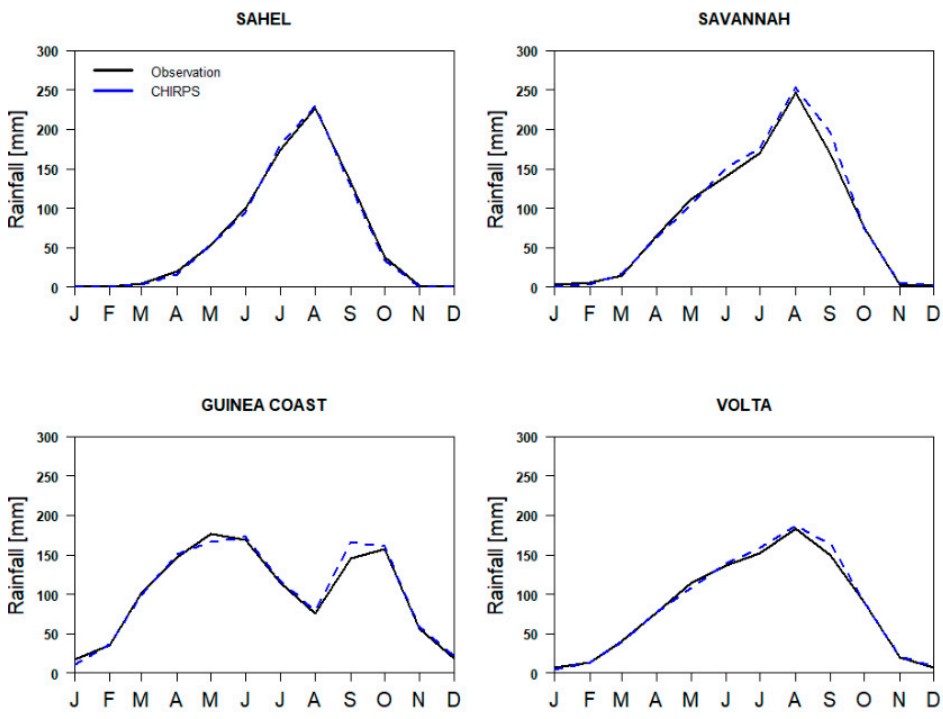

**Figure A1.** Validation of CHIRPS and NASA POWER data over the Volta Basin.

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
