# Peer review of "Rainfall Projections from Coupled Model Intercomparison Project Phase 6 in the Volta River Basin: Implications on Achieving Sustainable Development"

_sustainability, doi:10.3390/su15021472_

Round 1

Author Response

Responses to Reviwer #1 comments

S/N

COMMENTS

RESPONSE

1

More numerical results should be included in the abstract.

The abstract has been updated with additional numerical results. Kindly refer to the abstract in the revised version.

2

The keywords should be chosen differently from the title

The keywords have been updated with new keywords such as climate change; trend analysis, climate models; climate scenarios; Volta basin

3

The main objectives of this study and the novelty point should be discussed clearly

and in detail

Thank you for the comment.

The introduction has been restructured as recommended. Kindly refer to the introduction for the revised version of the revised manuscript.

4

The flowchart should be added for easier understanding for the readers.

A flowchart has been added. Figure 2 in the revised manuscript.

5

Why did you use the inverse distance weighting (IDW) technique for the spatial

distribution of changes in past and future rainfall? Why not Kriging? Please justify

it?

The choice of the IDW compared to other interpolation techniques is its usefulness when the distribution of the estimated parameters is not a normal distribution (Feng-Wen and Chen-Wuing, 2012) as in this study. The deterministic IDW interpolation technique has also been demonstrated to perform well in spatial rainfall distribution [Yiran, and Stringer, 2016; Larbi et al. 2018]. Detailed information on IDW is found in the study by Feng-Wen and Chen-Wuing, (2012).

Kindly refer to the methodology for the revised version in the revised manuscript.

6

I could not find detailed information about the IDW technique in the article.

Thank you for the observation. Additional information on IDW has been provided in the manuscript and it reads:

The IDW interpolation technique is based on a concept of distance weighting used to estimate the unknown spatial rainfall data from the known data of sites that are adjacent to the unknown site (Li and Heap 2008). Detailed information on IDW is found in the study by Feng-Wen and Chen-Wuing, (2012).

7

Which software was used to analyze the spatial distribution of changes in past and
future rainfall?

The spatial distributions of the changes in the past and future rainfall were also analyzed with the R-software.

8

Improve the quality of the Figure 3. Please add the x axis title (normalized

standard deviation)

Thank you for your comment. The quality of the figure can be attributed to the merging of four different figures. In the case of the x-axis, the output map does not give such an option so the suggestion made has rather been added to the title of the figure.

9

Limitations are not included in the present form.

The limitations of the study have now been included. It reads:  One limitation of this study is that, primarily for rainfall projections, it is preferred that a higher resolution regional climate model (RCMs) be employed. However, for this study, GCMs were used due to the lack of RCMs for CMIP 6.

10

Future directions are not clearly indicated

future studies should consider using RCMs or more than three GCMs for rainfall projections over the Area. In using GCMs also, it is recommended that future studies consider using the ensemble mean from more GCMs help reduce models projections' uncertainty and improve projections.

11

Conclusions should be thoroughly revised. Be clear and concise. Some sentences

are redundant. Summarize the main points

Thank you for the comment. The conclusion part has been revised with redundant sentences ben removed.

12

The text contains some occasional grammatical problems. It may need the

attention of someone fluent in English language to enhance the readability.

The authors have read through the document once again to check for grammar and language issues.

Reviewer 2 Report

See in the below.

Author Response

Responses to Reviwer #2 comments

S/N

COMMENTS

RESPONSE

1

But I cannot decide to accept this manuscript
due to the innovation points. Because in the introduction, the authors said the previous study
was limited to three countries or the performance of HRMIP simulation. In my opinion, this
is not important. You can try to search for the African research or even global research to
check if there is some similar studies about rainfall projections from SSP2-4.5 and SSP5-
8.5 of Coupled Model Intercomparison Project Phase 6. I believed that there should be some
relevant studies. Then what is the gap between your study and their studies? What problems
did you want to solve? Do you improve some methods or some parts? Due to this reason, I
cannot accept this paper. The major revision is suggested.

Thank you for your observation and comments. The introduction section has been revised to include some relevant studies. The problem statement and research gap have also been restructured.

Please kindly refer to the revised manuscript for the revised version.

2

In line 196, you said NorESM2-MM overestimates the amount of rainfall and MPIESM1-2-HR, BCC-CSM2-MR and the ensemble underestimates the rainfall values. It was suggested to explain the reason in the section of discussion. The similar suggestion is in line 231.

Thank you for your comment. The reason for differences in the models' performances and projections has been included in the discussion part of the manuscript. It reads:  According to Sood and Smakhtin 2015; Jain et al. 2019, the causes of uncertainties that lead to either overestimation or underestimation of values of GCMs can be mainly attributed to (i) approximations during numerical modeling, (ii) different feedback mechanisms (cloud and solar radiation, greenhouse gases, aerosols, natural and anthropogenic sources, ocean circulation, water vapor and warming, ice and snow albedo), and (iii) physical parameterizations, initializations, and model structures. In the Volta basin, as further posited by Agyekum et al. [29], the rainfall is determined mainly by two air masses namely: the dry cold “northeasterly” winds from the Sahara, and the moist warm “southwesterly” winds from the Atlantic Ocean, is likely to be overestimated or underestimated by GCMs that do not adequately represent these two air masses but exaggerate their effects.

3

In Figure 6, you compare the spatial distribution diagrams of different scenarios. It was

suggested to draw the difference diagrams between some scenario and the simulated

historical rainfall.

An additional map showing changes in rainfall between the scenarios and simulated historical has been added to figure 6 which now figures 7.

4

In line 368, you said that the projected increase in the mean annual rainfall in the Sahel

and Savannah zones can significantly improve agricultural productivity. In my opinion, if

there are more floods, the mean rainfall would also increase, but the productivity would

decrease. Therefore, it was suggested to consider the impact of flood.

The sentence has been revised and also the suggestion made has also been integrated. It reads:  under the SSP2-4.5 scenario, the projected increase in the mean annual rainfall (which is less than 10%) in the Sahel and Savannah zones can significantly improve agricultural productivity (i.e. increase in the crop, livestock, and fish yields) if well-distributed in the growing season thereby ensuring food security (i.e. SDG 2) and reduce poverty (i.e. SDG 1) in these areas by improving the living conditions of farming households. However, the very high increase in rainfall under the SSP5-8.5 scenario if not well distributed in the growing season may lead to flooding which can cause devastating effects on agriculture.

5

In line 442, it was suggested to add the limitation in this study and some future studies.

The limitations of the study and future studies have now been included. It reads:  One limitation of this study is that, primarily for rainfall projections, it is preferred that a higher resolution regional climate models (RCMs) be employed. However, for this study, GCMs were used due to a lack for RCMs for CMIP 6. future studies should consider using RCMs or more than three GCMs for rainfall projections over the Area. In using GCMs also, it is recommended that future studies consider using the ensemble mean from more GCMs helps reduce model projections uncertainty and improve projections.

Reviewer 3 Report

The work done by the authors is quite praiseworthy but there are numerous language issues and some technical lacking. After the critical observation, I found that there is a learning difficulty issue especially in understanding the results which made me suggest “MAJOR REVISION” for this article. Here I suggest authors work on the results critically: concise the information (consider the data which is significant and highlights of the study) and make the statements crystal clear. Discussion part is Ok (however, it may be improved after revising the results section) but the conclusion needs some fine-tuning with the consideration of the effective outcomes (highlight the novelty and how these projections can benefit in the climate change adaptation measures) and recommendation of the study. The future scope needs to be included at the end of the conclusion.

Though I found some difficulty while reading this article, I agree that this research has a great impact on planning and decision-making for the stakeholders in designing climate change adaptation.

My further comments are as given below.

Line: 25- Rainfall, Climate

Line: 45- (IPCC)

Line: 73 Bring “The aim” in next paragraph

Line: 82- provide the GPS coordinates in the standard style “Degrees Minutes Seconds (DMS)”

Line: 83- km2- superscript

Line: 89- North South- capitalize

Line: 90- 25.0oC (South) to 30.0oC (north)- Correct it. Degree original symbol 

Line: 122- equilibrium

Line: 124- carbon dioxide (CO2)

Line: 126- shared socioeconomic pathways

Line: 211- from the Figure 4

Line: 225- you may delete “for (a) Sahel, (b) Savannah, (c) Guinea Coast, and (d) the Volta basin”.

Line: annual rainfall (mm)

Line: 240- delete “Table 2 and Figure 5 show mean annual and monthly rainfall projections from the CMIP 240 6 models’ ensemble over the various zones and the Volta basin for the CMIP 6 historical 241 (1985-2014) and future (2021-2050) under SSP2-4.5 and SSP5-8.5 climate scenarios.” Fit the Table 2 and Figure 5 in the running texts accordingly.

Line: 253- Mean annual

Line: 272- projected rainfall (mm)

Line: 275- delete “Presented in Table 3 is” Start with “The trend”

Line: 276- climate change (Table 3).

Line: 284- delete “Presented in Figures 7 and Figure 8 are Violin plots showing the distributional character-284 istics of the projected changes in the annual mean rainfall by three (3) different GCMs and 285 their ensemble mean under SSP2-4.5 and SSP5-8.5 scenarios for Sahel, Savannah, Coastal 286 and the entire Volta basin respectively.” Fit the Figures 7 and Figure 8 in the running texts accordingly. Like in Line: 287- It is evident from the Figures 7 and Figure 8. Delete two Figures

Line: 323- The present study assessed

Line: 439-441- This is not the conclusion from your research. Remove it. End the conclusion with the future scope of your study.

Line: 531- Missing the reference here. Check!!!

Figure 3: Remove abcd from the figure as you are already naming it.

Table 2: Remove % given in bracket as it is detailed in the footnote

Language is a serious issue in this article. Seriously look into the grammatical and typo errors. 

Author Response

Responses to Reviwer #3 comments

S/N

COMMENTS

RESPONSE

1

There are numerous language issues and some technical lacking.

The authors have read through the document once again to check for grammar and language issues.

2

I suggest authors work on the results critically: concise the information (consider the data which is significant and highlights of the study) and make the statements crystal clear

The results section has been worked on.

3

Discussion part is Ok (however, it may be improved after revising the results section)

Thank you for your comment. Other reviewers' comments have been used to improve upon the discussion.

4

Conclusion needs some fine-tuning with the consideration of the effective outcomes (highlight the novelty and how these projections can benefit in the climate change adaptation measures) and recommendation of the study.

Thank you for the comment. The conclusion part has been revised with redundant sentences ben removed. A statement of recommendations has also been provided.

5

The future scope needs to be included at the end of the conclusion

The future scope has been added to the end of the conclusion. It reads:

future studies should consider using RCMs or more than three GCMs for rainfall projections over the Area. In using GCMs also, it is recommended that future studies consider using the ensemble mean from more GCMs help reduce models projections' uncertainty and improve projections.

6

Line: 45- (IPCC)

Line: 73 Bring “The aim” in next paragraph

All comments are duly addressed in the manuscript

7

Line: 82- provide the GPS coordinates in the standard style “Degrees Minutes Seconds (DMS)”

longitudes 5∘30’30’’W to 2∘0’30’’E, and latitudes 6∘0’30’’ to 15∘0’30’’N

8

Line: 83- km2- superscript

Line: 89- North South- capitalize

Line: 90- 25.0oC (South) to 30.0oC (north)- Correct it. Degree original symbol 

Line: 122- equilibrium

Line: 124- carbon dioxide (CO2)

Line: 126- shared socioeconomic pathways

Line: 211- from the Figure 4

Line: 225- you may delete “for (a) Sahel, (b) Savannah, (c) Guinea Coast, and (d) the Volta basin”.

Line: annual rainfall (mm)

All comments are duly addressed in the manuscript

9

Line: 240- delete “Table 2 and Figure 5 show mean annual and monthly rainfall projections from the CMIP 240 6 models’ ensemble over the various zones and the Volta basin for the CMIP 6 historical 241 (1985-2014) and future (2021-2050) under SSP2-4.5 and SSP5-8.5 climate scenarios.” Fit the Table 2 and Figure 5 in the running texts accordingly.

They are all deleted. It can be found in the manuscript.

10

Line: 253- Mean annual

Line: 272- projected rainfall (mm)

Line: 275- delete “Presented in Table 3 is” Start with “The trend”

Line: 276- climate change (Table 3).

All comments are duly addressed in the manuscript

11

Line: 284- delete “Presented in Figures 7 and Figure 8 are Violin plots showing the distributional character-284 istics of the projected changes in the annual mean rainfall by three (3) different GCMs and 285 their ensemble mean under SSP2-4.5 and SSP5-8.5 scenarios for Sahel, Savannah, Coastal 286 and the entire Volta basin respectively.” Fit the Figures 7 and Figure 8 in the running texts accordingly. Like in Line: 287- It is evident from the Figures 7 and Figure 8. Delete two Figures

They are all deleted. It can be found in the manuscript.

12

Line: 323- The present study assessed

This is corrected.

13

Line: 439-441- This is not the conclusion from your research. Remove it. End the conclusion with the future scope of your study.

Line: 439-441 has been removed.

The future scope has also been added. It reads:

future studies should consider using RCMs or more than three GCMs for rainfall projections over the Area. In using GCMs also, it is recommended that future studies consider using the ensemble mean from more GCMs helps reduce models projections' uncertainty and improve projections.

14

Line: 531- Missing the reference here. Check!!!

Figure 3: Remove abcd from the figure as you are already naming it.

Table 2: Remove % given in bracket as it is detailed in the footnote

All comments are duly addressed in the manuscript.

15

Language is a serious issue in this article. Seriously look into the grammatical and typo errors.

The authors have read through the document once again to check for grammar and language issues.

Round 2

Reviewer 1 Report

The authors carefully addressed all of the comments during the revision process. The paper is now acceptable.

Author Response

Thank you for your positive feedback. We are very grateful.

Reviewer 2 Report

The authors have revised the manuscript well. But in line 97, it reads as "These studies, however, are rarely brought up or positioned in the context of sustainable development discourse. " I cannot understand this sentence. I hope that the authors can give more explanation here. Others are fine.

Author Response

Comment:  The authors have revised the manuscript well. But in line 97, it reads as “these studies, however, are rarely brought up or positioned in the context of sustainable development discourse”. I cannot understand this sentence. I hope that the authors can give more explanation here.

Response: Thank you for your positive feedback and also the comment. The statement has been reconstructed. It reads: “However, studies on climate change modeling are not often discussed within the context of sustainable development, especially how such studies can help to achieve the sustainable development goals. This study, therefore, seeks to move the discussion of the findings within the sustainable development discourse.

Reviewer 3 Report

The authors worked on the manuscript and technically improved the draft in light of my comments. I endorse this article for publication. 

Author Response

(The authors gave the same response as above.)
